# MitoScape: A big-data, machine-learning platform for obtaining mitochondrial DNA from next-generation sequencing data

Larry N. Singh[1]*, Brian Ennis[2], Bryn Loneragan[3], Noah L. Tsao[4], M. Isabel G. Lopez Sanchez[3], Jianping Li[5], Patrick Acheampong[1], Oanh Tran[6], Ian A. Trounce[3], Yuankun Zhu[2], Prasanth Potluri[1], Regeneron Genetics Center[7], Beverly S. Emanuel[6], Daniel J. Rader[8], Zoltan Arany[9], Scott M. Damrauer[4], Adam C. Resnick[2], Stewart A. Anderson[5], Douglas C. Wallace[1]

1 Center for Mitochondrial and Epigenomic Medicine, Division of Human Genetics, The Children's Hospital of Philadelphia, Philadelphia, Pennsylvania, United States of America, 2 Center for Data-Driven Discovery in Biomedicine (D3b), The Children's Hospital of Philadelphia, Philadelphia, Pennsylvania, United States of America, 3 Center for Eye Research Australia, Ophthalmology, Department of Surgery, University of Melbourne, Melbourne, Australia, 4 Department of Surgery, Perelman School of Medicine, University of Pennsylvania, Philadelphia, Pennsylvania, United States of America, 5 Department of Psychiatry, The Children's Hospital of Philadelphia and the University of Pennsylvania, Philadelphia, Pennsylvania, United States of America, 6 22q and You Center, Division of Human Genetics, The Children's Hospital of Philadelphia and the University of Pennsylvania, Philadelphia, Pennsylvania, United States of America, 7 Regeneron Genetics Center, Tarrytown, New York, United States of America, 8 Department of Genetics, Perelman School of Medicine, University of Pennsylvania, Philadelphia, Pennsylvania, United States of America, 9 Cardiovascular Institute, Perelman School of Medicine, University of Pennsylvania, Philadelphia, Pennsylvania, United States of America

* larrynsingh@gmail.com

**Data Availability Statement:** Data specific to HCM analysis are available from the Penn Medicine Biobank (https://pmbb.med.upenn.edu). All other data, including Benchmark data, are available via

## Abstract

The growing number of next-generation sequencing (NGS) data presents a unique opportunity to study the combined impact of mitochondrial and nuclear-encoded genetic variation in complex disease. Mitochondrial DNA variants and in particular, heteroplasmic variants, are critical for determining human disease severity. While there are approaches for obtaining mitochondrial DNA variants from NGS data, these software do not account for the unique characteristics of mitochondrial genetics and can be inaccurate even for homoplasmic variants. We introduce MitoScape, a novel, big-data, software for extracting mitochondrial DNA sequences from NGS. MitoScape adopts a novel departure from other algorithms by using machine learning to model the unique characteristics of mitochondrial genetics. We also employ a novel approach of using rho-zero (mitochondrial DNA-depleted) data to model nuclear-encoded mitochondrial sequences. We showed that MitoScape produces accurate heteroplasmy estimates using gold-standard mitochondrial DNA data. We provide a comprehensive comparison of the most common tools for obtaining mtDNA variants from NGS and showed that MitoScape had superior performance to compared tools in every statistically category we compared, including false positives and false negatives. By applying MitoScape to common disease examples, we illustrate how MitoScape facilitates important heteroplasmy-disease association discoveries by expanding upon a reported association between hypertrophic cardiomyopathy and mitochondrial haplogroup T in men (adjusted p-

authorized access from https://cavatica.sbgenomics.com/u/cavatica/22q11-deletion-syndrome-project/.

**Funding:** This work was supported by grants awarded to SA (National Institutes of Mental Health - MH110185) and DCW (National Institute of Neurological Disorders and Stroke: NS021328, National Institutes of Mental Health: MH108592, and Office of the Director: OD010944). The funders had no role in study design, data collection and analysis, decision to publish, or preparation of the manuscript.

**Competing interests:** The authors have declared that no competing interests exist.

value = 0.003). The improved accuracy of mitochondrial DNA variants produced by MitoScape will be instrumental in diagnosing disease in the context of personalized medicine and clinical diagnostics.

## Author summary

Recent studies have highlighted the importance of mitochondrial DNA variation in both primary mitochondrial disease and complex, human pathology including COVID-19, and space-flight stress. The vast amount of existing, next-generation sequencing (NGS) data can be leveraged to interrogate both nuclear and mitochondrial DNA (mtDNA) sequence simultaneously, allowing for analysis of the interplay between mitochondrial and nuclear encoded genes in mitochondrial function. Identifying mtDNA sequence accurately is complicated by the presence of nuclear encoded mitochondrial sequences (NUMTs), which are homologous to mtDNA. Current software for analyzing mtDNA from NGS do not accurately model the unique characteristics of mitochondrial genetics. We introduce MitoScape, a novel, big-data, software which models mitochondrial genetics through machine learning to accurately identify mtDNA sequence from NGS data. MitoScape takes advantage of rho-zero cell data to model the characteristics of NUMTs. We show that MitoScape produces more accurate heteroplasmy estimates compared to published software. We provide an example of applying MitoScape in replicating an association between hypertrophic cardiomyopathy and mitochondrial haplogroup T in men. MitoScape is an important contribution to mitochondrial genomics allowing for accurate mtDNA variants, and the ability to tailor mtDNA analysis in different population and disease contexts, which is not available in other software.

This is a *PLOS Computational Biology* Software paper.

## Introduction

Both mitochondrial DNA (mtDNA) and nuclear DNA (nDNA) variants are known to impair the function and structure of mitochondria, leading to *primary mitochondrial disease* [1]. But studies have also implicated mtDNA variants in a myriad of common, complex, human diseases, including cancer, cardiovascular disease, diabetes and neurodegenerative disease [2–6]. More recently, mtDNA variation and damage have been implicated in COVID-19 [7,8] and even spaceflight [9]. Thus, there is a need to interrogate both mtDNA and nDNA variants simultaneously in both primary mitochondrial and complex disease. Existing, large-scale, next-generation sequencing (NGS) datasets are a valuable resource for retrospectively analyzing both mtDNA and nDNA variation in an array of common diseases. Today, such large datasets are both abundant and necessary in genetic association studies for overcoming biases and false negatives due to a lack of statistical power. For example, the Cancer Mitochondrial Atlas (TCMA) identified signatures of mtDNA variation in different forms of cancer, using data from thousands of whole genome sequencing (WGS) samples [5].

Due to fundamental differences between Mendelian and mitochondrial genetics, erroneous interpretation and poor data analysis are common in analyzing mtDNA [10]. Inherent

complexities of mitochondrial genetics include heteroplasmy, nuclear-encoded mtDNA sequences (NUMTs), and low-complexity regions. *Heteroplasmy* is the presence of multiple copies of mtDNA with differing nucleotide sequences in a single cell or population of cells and tissues. Heteroplasmy arises since each human nucleated cell typically comprises 100s to 1000s of mitochondria with each mitochondrion containing 2–8 copies of mtDNA, and each copy accumulating independent variants. Low and high percentages of mtDNA variants give rise to low and high heteroplasmy, respectively. While nDNA variants follow the laws of Mendelian genetics, mtDNA variants abide by the principles of population genetics [6]. The prevailing hypothesis is that cells are resilient to low levels of mtDNA having genetic defects, and bio-chemical defects only occur once the levels of defective mtDNA exceed a critical threshold, a phenomenon termed the *threshold effect* [11]. Since the level of heteroplasmy appears to be positively correlated with disease severity, the threshold effect suggests that high percentages of an mtDNA variant are required to produce a functional effect. But detecting low-level het-eroplasmy is essential for two reasons: first, since heteroplasmy varies by tissue, low hetero-plasmy in blood may allude to high heteroplasmy in internal organs; and second, low-level heteroplasmy variants appear to be widespread and present in all humans, and can be heritable and functional [12]. Low-level heteroplasmy also increases with age and hence, may contribute to common late-onset diseases [12]. Consequently, the primary question in mitochondrial genetics, is not whether or not a variant exists, but at what heteroplasmy level. Thus, accurate computation of mtDNA heteroplasmy especially at low levels, is crucial for understanding and diagnosing complex disease.

Contributing to the complexity of obtaining heteroplasmy levels, are NUMTs, which result from mtDNA fragments that were transferred into the nucleus and incorporated into the nuclear genome [13]. The formation of NUMTs, termed *numtogenesis*, is a dynamic, on-going, and evolutionarily-conserved process [14]. Human NUMTs range from 64–100% sequence identity to mtDNA and vary in length from approximately 40bp to almost the entire mtDNA [15]. Reads from standard NGS are shorter than many NUMTs, which means that using sequence alignments alone to distinguish mtDNA from NUMTs is prone to error. Align-ment of a NUMT to the mtDNA, results in a false positive and inflation of heteroplasmy. Con-versely, alignment of mtDNA to NUMTs, results in false negatives and an underestimate of heteroplasmy. The effects of NUMT variants are often underappreciated in studies of mtDNA heteroplasmy [16], and the effect of NUMTs on mtDNA heteroplasmy is perceived to be negli-gible. This assumption is flawed, however, since the human genome contains over 700 germ-line NUMTs and multiple NUMTs correspond to the same mtDNA region [17].

Current, computational methods for obtaining mtDNA variants from NGS data can be broadly classified into two categories: 1) those that rely on unique alignments of mtDNA (unique alignment approach), and 2) those that rely on post-filtering of mtDNA variants (post-filtering approach). Unique alignment approaches such as MToolBox [18], contend with NUMTs by discarding sequence reads that do not uniquely map to mtDNA. Such approaches that solely rely on sequence alignment are likely to result in genuine mtDNA being discarded and both over- and underestimates of heteroplasmy [6,19,20]. Post-filtering approaches such as mtDNA-Server [21], limit the influence of NUMTs on mtDNA variants by discarding or flagging variants that flank published NUMT regions [5]. Post-filtering has several drawbacks compared to the unique alignment approach. First, filtering can result in potentially important variants being overlooked. Second, the composition of NUMTs varies depending on the dis-ease, population and even individual [15,22], making a complete list of NUMT regions impractical to formulate. Furthermore, the NUMTs can also vary by tissue or cells collected, meaning that post-filtering is not generalizable to different samples collected. Third, post-fil-tering of mtDNA variants suspected to be from NUMTs overlooks the fact that mtDNA

variants are often in high linkage-disequilibrium and cannot be treated independently of each other. Post-filtering loses information about which variants occur on the same read, an important factor in determining legitimate mtDNA variants. Fourth, mtDNA copy number estimation, another important measure of mtDNA variability, is not possible from post-filtering methods. Fifth, post-filtering does not accommodate retrospective quality-control analysis of NGS reads. Sixth, from a software engineering perspective, post-filtering systems are tightly coupled, inflexible to changes in individual components, for example, changes to the alignment software, and do not scale or generalize well. Therefore, in summary, accurate mtDNA variants are only possible from accurate mtDNA alignments. Low complexity regions suffer similar consequences as NUMTs, as nucleotides that flank these regions are commonly filtered [21,23].

Published methods for obtaining mtDNA variants are based on some combination of the aforementioned techniques and assumptions. These approaches rely on rigid and often arbitrary thresholds for filtering NUMTs, and do not adequately model the variable nature of mtDNA. Sequence alignment alone is insufficient to distinguish mtDNA from NUMTs, and filtering of NUMT and low-complexity regions is restrictive. We present *MitoScape*, a novel, machine learning-based software to align mtDNA from NGS data (Fig 1 and Design and Implementation). MitoScape incorporates two novel advancements: first, we use machine learning to model and learn the unique characteristics of mtDNA and NUMTs; and second, we use rho-zero cells for the first time as a source of NUMTs for training the classifier. An advantage of a machine learning classifier is that mtDNA sequence alignments are assigned probabilities of being from mtDNA as opposed to being discarded, unknown to the user. Furthermore, we take advantage of the fact that machine learning can learn the salient characteristics of NGS to discriminate mtDNA reads from NUMT reads, without making unnecessary and arbitrary assumptions. The training datasets can also be altered to accommodate different populations or diseases, for example. MitoScape is a big-data, cloud-based software system and is scalable to virtually any number of NGS samples. The main application of MitoScape is to retrospectively analyze mtDNA variation in existing NGS data in common disease contexts. We tested MitoScape on a novel, gold-standard benchmark dataset comprised of mtDNA-enriched data.

## Design and implementation

### Ethics statement

The IBBC study was approved by "The Committees for the Protection of Human Subjects (IRB)" at the Children's Hospital of Philadelphia, under protocol "Genetic Modifiers of 22q11.2 Abnormalities", IRB 07–005352. Each participant and his or her caregiver, when appropriate, provided informed written consent/assent to participate prior to recruitment.

### Training data

Aligning NGS reads to the human genome reference sequence results in a subset of reads that align ambiguously to both the mtDNA (revised Cambridge Reference Sequence or rCRS [24]) and the nDNA (Fig 1). Sequence alignments alone cannot discriminate mtDNA from NUMTs. Rather than rely on restrictive filters that use hard thresholds, we developed a novel approach using a machine learning classifier to compute the probability that a sequencing read is from mtDNA. Our classifier automatically learns characteristics of both mtDNA and NUMT reads to better align mtDNA sequences. We use a positive training set comprised of authentic mtDNA reads, and a negative training set comprised of NUMT sequences for supervised learning. For the negative training set, we sequenced both wild-type and mtDNA-

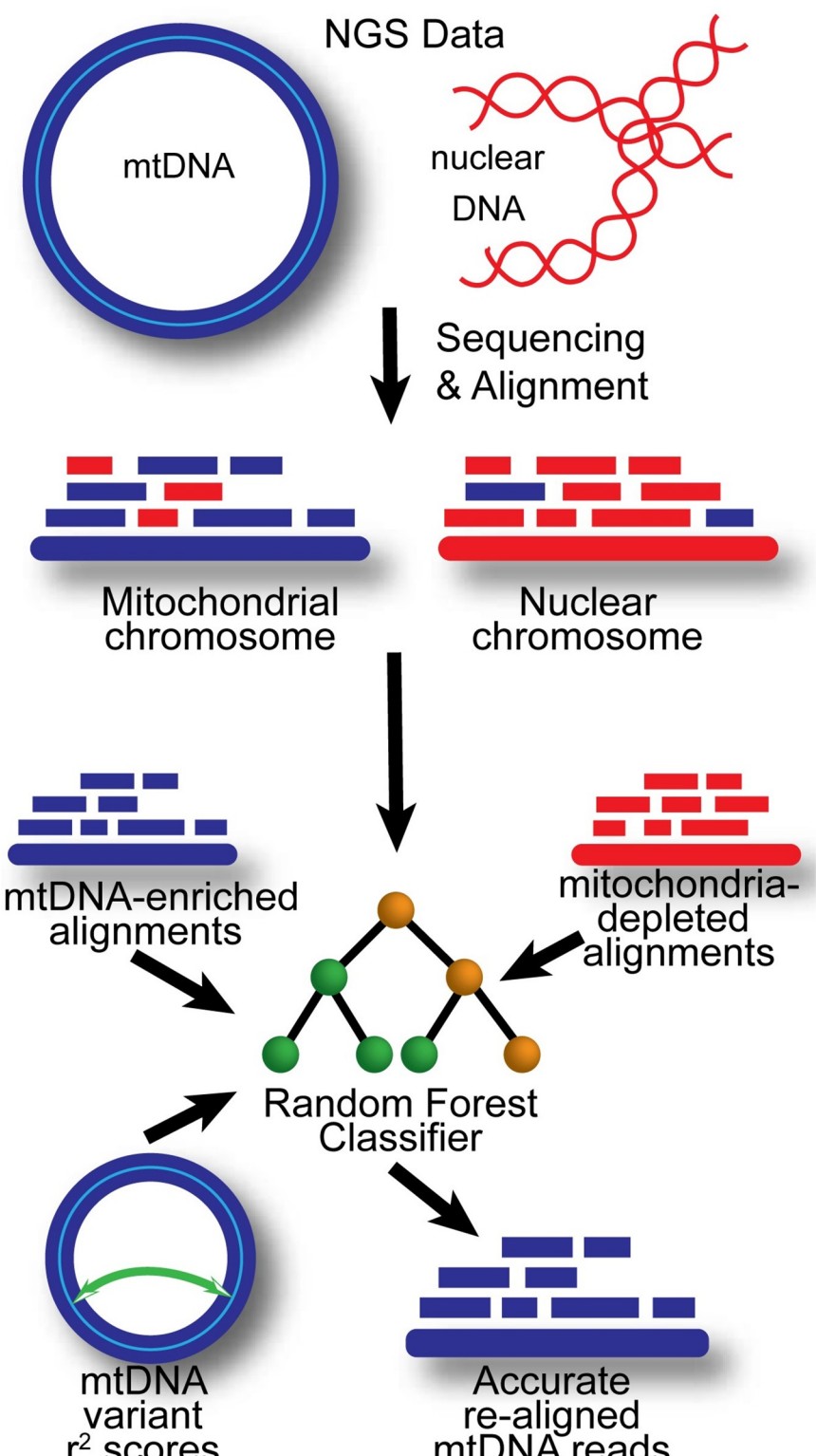

**Fig 1. Overview of MitoScape algorithm.** WGS data containing total DNA includes both mtDNA and NUMTs. After alignment to the reference genome, some NUMTs will erroneously align to mtDNA, and some mtDNA will erroneously align to NUMTs. To correct these alignment errors, we use a random forest classifier. The classifier is trained on positive, mtDNA-enriched alignments, and negative mitochondria-depleted alignments. We also use linkage disequilibrium r2 scores and common NUMT locations to determine the probability that an ambiguous read is truly from mtDNA.

depleted (rho-zero or $\rho^0$) WAL2A lymphoblastoid and 143B osteosarcoma cell lines [25], each in duplicate, for a total of eight samples. The use of rho-zero cells to model NGS characteristics of NUMT sequences is a novel departure from other computational approaches for aligning mtDNA. To obtain NUMT sequences from each sample:

1. Align reads to the rCRS.

2. Re-align the aligned reads from step 1 to the nuclear genome, i.e. all of the human reference genome except for the rCRS.

3. These aligned reads comprise the negative training set of NUMTs.

For the positive training set, we sequenced ten samples of mtDNA-enriched samples generated by amplifying mtDNA in two overlapping long-range PCR fragments of about 8,500bp each from WAL2A cell lines. The resulting amplified DNA sequences were then aligned to the rCRS to create our positive training set of mtDNA reads.

## Feature selection and machine learning classifier

We chose a random forest classifier [26] for resolving ambiguities in aligning reads to the rCRS, due to this classifier's simplicity and resistance to overfitting when the number of training samples is small, as is our case. We also tested gradient boosted trees as a classifier but found that random forest classifiers had a lower test error. Training of the classifier was performed using k-fold cross-validation whereby 80% of all reads from all samples were chosen at random for training and the remaining 20% used for validation. An 80%-20% split corresponds to k = 5, with four groups for training and one group for testing. Defined as 2 * (precision−recall) / (precision + recall), the resulting F1 score from k-fold cross-validation was 0.81.

To train the classifier, we required quantities that are measurable and informative of whether a read is a NUMT or mtDNA, termed *features*. Several features were tested for model selection using the random forest classifier (**Table 1**). We discuss those features here. According to data in our human mitochondrial genome database, MITOMAP (www.mitomap.org), approximately 55% of mtDNA loci have been reported as mtDNA variants. Furthermore, the frequency of each mtDNA variant is dependent on the population of interest. Therefore,

**Table 1. Summary of features considered for random forest classifier.** Each feature is considered for determining whether the alignment of the read in SAM format corresponds to mtDNA or a NUMT. The SAM Tag field indicates the corresponding field in the SAM alignment format specification. Features in bold were used in the final model for the random forest classifier.

| Feature | Description | SAM Tag |
|---|---|---|
| mtDNA edit distance | Edit distance between the read and aligned mtDNA sequence. A lower edit distance indicates that there are fewer differences between the read sequence and the reference sequence. | NM |
| Nuclear edit distance | Edit distance between the read and aligned nuclear genomic sequence. | NM |
| mtDNA alignments | Number of alignments to the mtDNA sequence | NH |
| Nuclear alignments | Number of alignments to the nuclear genomic sequences | NH |
| Mapping score | Non-normalized mapping quality score | XQ |
| Mapping quality | Mapping quality field of SAM entry | |
| MT LD | Linkage disequilibrium scores of variants within the paired alignment to mtDNA | |
| NUMT Overlap | Percentage of overlap of the paired read with a known, validated NUMT region. | |

relying solely on mtDNA variant frequency for filtering is not informative and would lead to overfitting of the machine learning algorithm. Instead, we developed a novel solution as follows. We observe that due to the high linkage disequilibrium (LD) among mtDNA variants, many mtDNA variants are inherited as a haplotype or haplogroup [3]. Therefore, we computed LD $r^2$ scores for all mtDNA variants from 45,494 GenBank hand-curated mtDNA sequences from MITOMAP, using Plink version 1.9 (atgu.mgh.harvard.edu/plinkseq/). These LD scores were used to compute the probability of two variants on a paired-end read appearing together on the same mtDNA sequence. A low probability suggests that the read is from a NUMT rather than mtDNA. To obtain an initial set of variants in each read, we developed a basic variant caller in MitoScape, which called variants based on the mismatching positions (MD) tags in the sequence alignment/map format (SAM) fields of an aligned paired-end read.

The composition of NUMTs in a genome is highly variable and depends on population, disease, tissue, and even individual. Consequently, providing an exhaustive list of NUMTs is counter-productive and will lead to over-fitting of the classifier. In MitoScape, we allow the user to select a list of known NUMTs as a parameter to the software. We provide a generic list of common, experimentally validated NUMTs [17] based on the most common tissue used in NGS studies: blood. Using an input list of NUMTs, MitoScape calculates the fraction that an ambiguous paired-end read overlaps with a known NUMT region. This score, referred to as NUMT overlap (**Table 1**), is used as a feature. Based on variable or feature importance and model accuracy (S1 Fig), the final model used mtDNA edit distance, nuclear edit distance, mtDNA LD scores and NUMT overlap, using 128 trees in the random classifier to obtain the probability that an NGS read is from mtDNA.

The following is a summary of the workflow for calling mtDNA variants using MitoScape:

1. Align the WGS sample to the mtDNA reference sequence.

2. Re-align the aligned sequences from step 1 to the non-mtDNA reference sequence.

3. Call MitoScape with the outputs from steps 1 and 2 to classify ambiguous mtDNA reads.

4. Call variants on the output mtDNA sequences from step 3.

Several design choices were made to ensure that MitoScape employed a flexible and decoupled architecture. Every major software component can be replaced without changing any code. For example, gsnap [27] has the unique ability to align circular DNA such as the rCRS; however, another short-read sequence aligner could be used. For calling variants we utilized Mutect2, which was originally designed for calling cancer variants. All of the software developed was designed using Scala, a modern, scalable, functional and object-oriented programming language which runs on the Java Virtual Machine (JVM). For processing of the aligned reads, we used ADAM version 0.32.0 [28,29], a library designed for big-data, genomic analysis using Apache Spark. For machine learning paradigms, Apache Spark version 3.0.0, a fast, unified analytics engine for big-data processing was chosen. Apache Spark also improves on many of the shortcomings of Apache Hadoop, including improved performance and flexibility.

## Customization of MitoScape

The performance of a machine learning classifier is dependent on the quality of the training data. MitoScape was designed to be scalable and flexible, and so it is a trivial matter to add more or different training data to the model—a feature that is unique to MitoScape and not present in other tools. For instance, if studying cancer, a training data comprised of cancer samples would be more appropriate. The majority of NGS studies contain lymphocyte DNA and hence, the lymphoblastoid cell lines are a suitable model. Similarly, both the list of NUMT

scores and the linkage disequilibrium scores could be customized by adding more data samples or, for example, restricting samples based on population or tissue. The performance of MitoScape will improve as more data is generated and analyzed, and the system is designed to accommodate more data. The ability to select mtDNA variants based on sensitivity and specificity via the prediction probability parameter of MitoScape is another useful enhancement that is unavailable in other software.

## Results

### Benchmark dataset

The current gold standard used in clinical genetic labs for obtaining mtDNA variants is long-range PCR amplification of mtDNA sequencing followed by NGS and variant calling [30]. Furthermore, since the vast majority of existing NGS is from Illumina paired-end reads, and MitoScape was designed to obtain mtDNA sequences from NGS, we tested MitoScape using this approach in the context of a complex disease: schizophrenia. We obtained nine blood samples from the 22q11IBBC study, which comprised subjects having 22q11.2 deletion syndrome (22q11DS) and schizophrenia [31]. These nine samples are completely different from the ten used as in the positive training set, and hence are an appropriate and valid test set. To obtain gold-standard mtDNA sequences, we amplified mtDNA from these samples in two long-range overlapping PCR fragments (S1 Supplementary Methods). The purity of amplified sequence varies with different PCR primers. Hence, we designed, developed and tested novel PCR primers for human blood cells (S1 Supplementary Methods and S2 and S3 Figs). We then sequenced the amplicons on an Illumina MiSeq sequencer using twice as long 2x300bp paired-end reads than the common 2x150bp reads to produce more accurate alignments. The reads were then aligned to the human genome reference sequence (GRCh38). To maximize accuracy of alignments, we adopted a stringent, conservative approach in which only reads having at least 270bp (>90% of the maximum read length) aligning to the rCRS were retained. At least 90% of all sequenced reads (median = 92%) from all nine samples aligned to the GRCh38. Of all the aligned reads, at least 87% (median = 90%) aligned to the rCRS (S2 and S3 Figs) from each sample. These resulting sequences represent pure mtDNA sequences, free of NUMTs, and comprises our "*Benchmark*" mtDNA sequences (Fig 2). Our benchmark mtDNA sequences offer many improvements compared to validation samples used in previously published software for NGS analysis of human mtDNA, such as mtDNA-Server [21] (S1 Supplementary Methods). Following standard practice for machine learning training, the Benchmark dataset is a completely different set of samples from the training datasets to reduce bias.

### Model testing and evaluation

MitoScape distills mtDNA sequences from NGS data. Our goal is to compare the mtDNA sequences from MitoScape to the Benchmark mtDNA. We measured the performance of MitoScape by comparing the heteroplasmy levels of variants from MitoScape mtDNA to the heteroplasmy levels of the same variants from the Benchmark mtDNA. To obtain variants from the aligned mtDNA sequences, we selected a cancer variant caller with the ability to handle the polyploid genome of mitochondria: Mutect2 v4.1.9.0 [32], with mitochondrial-mode set to true, followed by FilterMutectCalls with default options. Only variants from the Benchmark mtDNA sequences that passed FilterMutectCalls filtering were considered as part of our Benchmark mtDNA variant set. Note that this test dataset is completely independent of the training data, and hence is a legitimate test for performance. We refer to the mtDNA variants called from the Benchmark mtDNA sequences as the *Benchmark variants*.

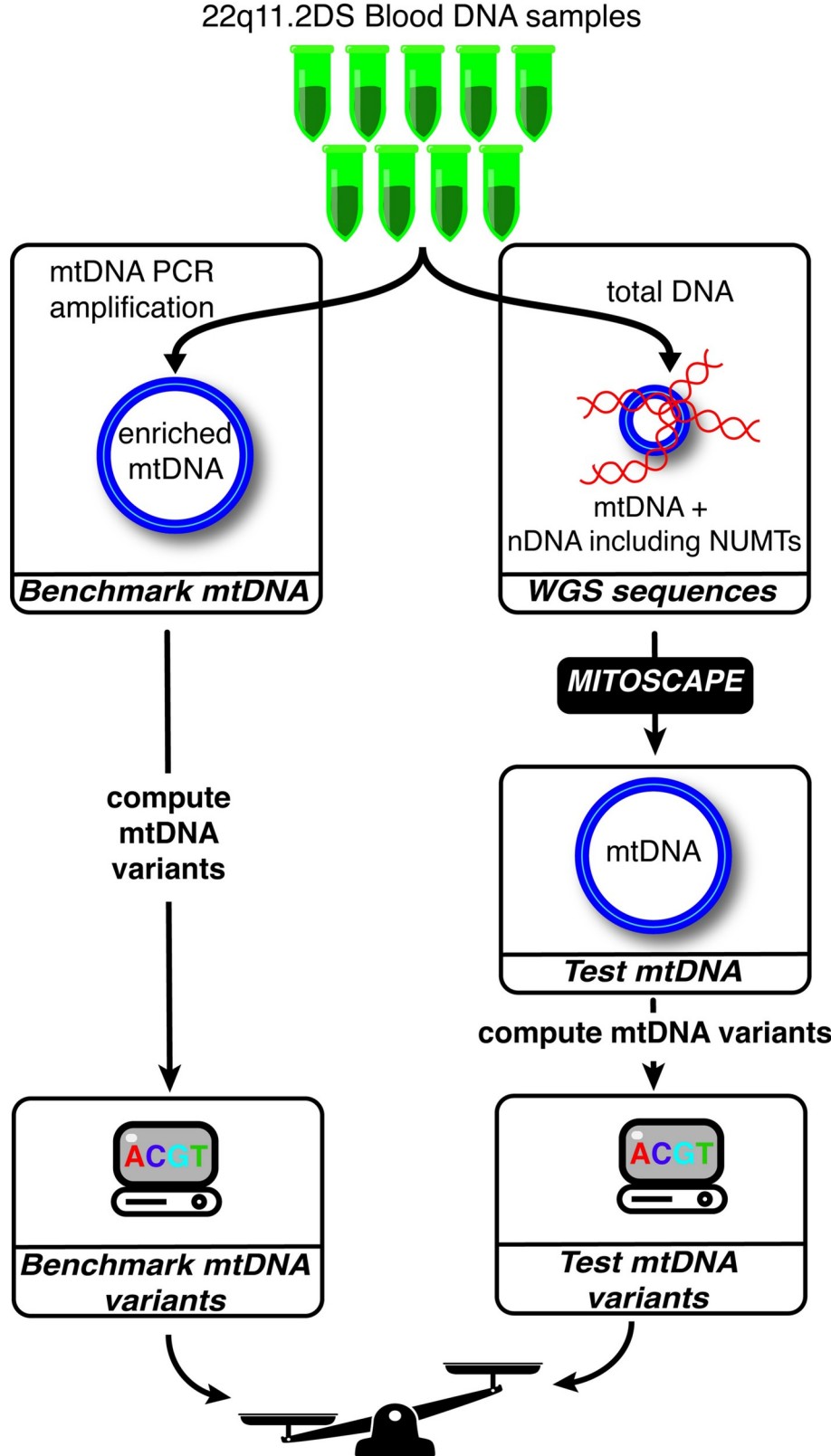

**Fig 2. Outline of testing scheme for MitoScape.** Nine different 22q11.2 deletion syndrome (DS) samples were chosen for performance testing. For each sample, we performed both 1) PCR amplification to enrich mtDNA, and 2) whole genome sequencing (WGS). MitoScape was applied to the WGS samples to obtain accurate mtDNA alignments. Variants were then called from both the resulting mtDNA from both mtDNA enrichment (Benchmark mtDNA) and WGS (test mtDNA) to obtain mtDNA variants. The Benchmark mtDNA variants represent the gold-standard variants from the nine samples. The test mtDNA variants were then compared to the Benchmark set for evaluation of the performance of MitoScape. Heteroplasmy values of the test mtDNA variants similar to those of the Benchmark variants, indicates that MitoScape is doing well, and vice-versa.

For the same nine 22q11DS samples used in the Benchmark mtDNA sequences, we performed WGS sequencing. WGS data are not enriched for mtDNA and hence also contain NUMTs. Therefore, MitoScape is required to discriminate mtDNA from NUMTs (Fig 2). We obtained mtDNA sequences from the WGS data using MitoScape with a prediction probability of 0.5 followed by Mutect2 to call variants. It is important to note that we do not use FilterMutectCalls in our tests of MitoScape, so we are not using Mutect2's filters. We then computed the difference in heteroplasmy levels between the Benchmark variants and the MitoScape mtDNA variants at each mtDNA variant locus for each sample. We defined *heteroplasmy error* of MitoScape in a given sample and specific mtDNA locus as the heteroplasmy in the Benchmark mtDNA variants minus the heteroplasmy computed using MitoScape i.e., heteroplasmy error = Benchmark heteroplasmy–MitoScape heteroplasmy. Hence, the closer the heteroplasmy error is to zero, the more the heteroplasmy results of MitoScape match the Benchmark heteroplasmy values. Positive heteroplasmy error indicates that the heteroplasmy estimate of a given variant was higher in the Benchmark variant set than that of the same variant in the MitoScape variant set. Therefore, increased positive heteroplasmy error suggests an increased chance of being a false negative. Any variants that were not called in a variant set are regarded as having heteroplasmy equal to 0.

## Overall performance of MitoScape

The Benchmark variants consisted of low-level heteroplasmy variants at almost every locus in the mtDNA, and thus, was a comprehensive test dataset (Figs 3 and S4). The nine test samples used here are substantially more than the two samples used for testing in mtDNA-server. Based on our variant calls we have captured low heteroplasmy variants across the entire mtDNA genome, thus adding substantially more samples is unlikely to add significantly more information. Since each test sample requires both mtDNA-enriched and (2x300 Illumina) WGS data, adding significantly more samples was also cost-prohibitive. MitoScape had heteroplasmy error of approximately zero for most variants except for two mtDNA regions at m.303 and m.16184 (Fig 3). Closer inspection reveals that these loci are within low-complexity regions consisting of homopolymer runs of almost exclusively cytosines. The low read depth of the variants in these two regions (Fig 3) and the high GC content render these two regions difficult to sequence and emphasizes that care should be taken when analyzing variants in these two regions. Due to sampling error, higher levels of heteroplasmy are likely to have higher variances than lower levels of heteroplasmy. Therefore, to account for differences in variances, we also scaled the heteroplasmy by $p(1-p)$, where $p$ is the benchmark heteroplasmy level. We have defined this measure as the *scaled heteroplasmy level*. The trends in heteroplasmy levels are similar between the raw heteroplasmy errors (Fig 3A) and the scaled heteroplasmy levels (Fig 3B).

The mean estimated heteroplasmy of variants from MitoScape was within 1% of that computed from the Benchmark. The standard deviation of heteroplasmy error approached zero as the read depth of the variant increased (Fig 4), emphasizing the importance of read depth in

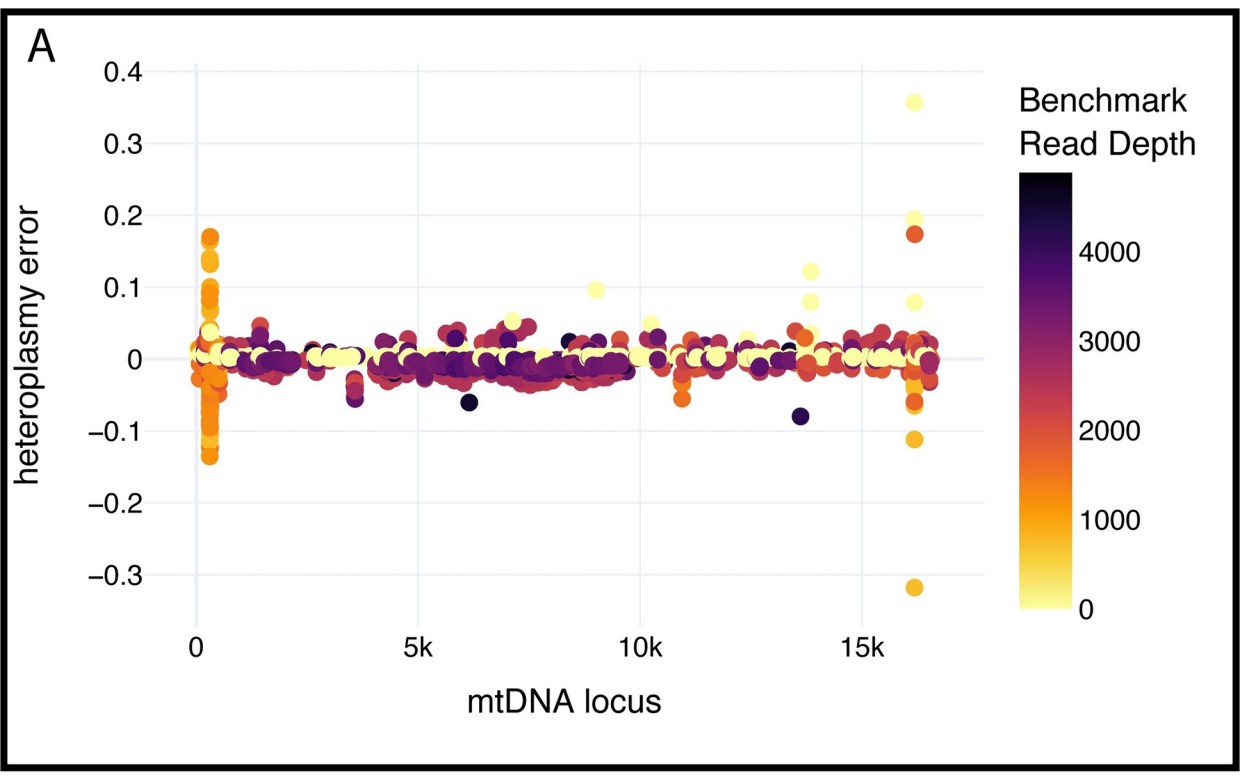

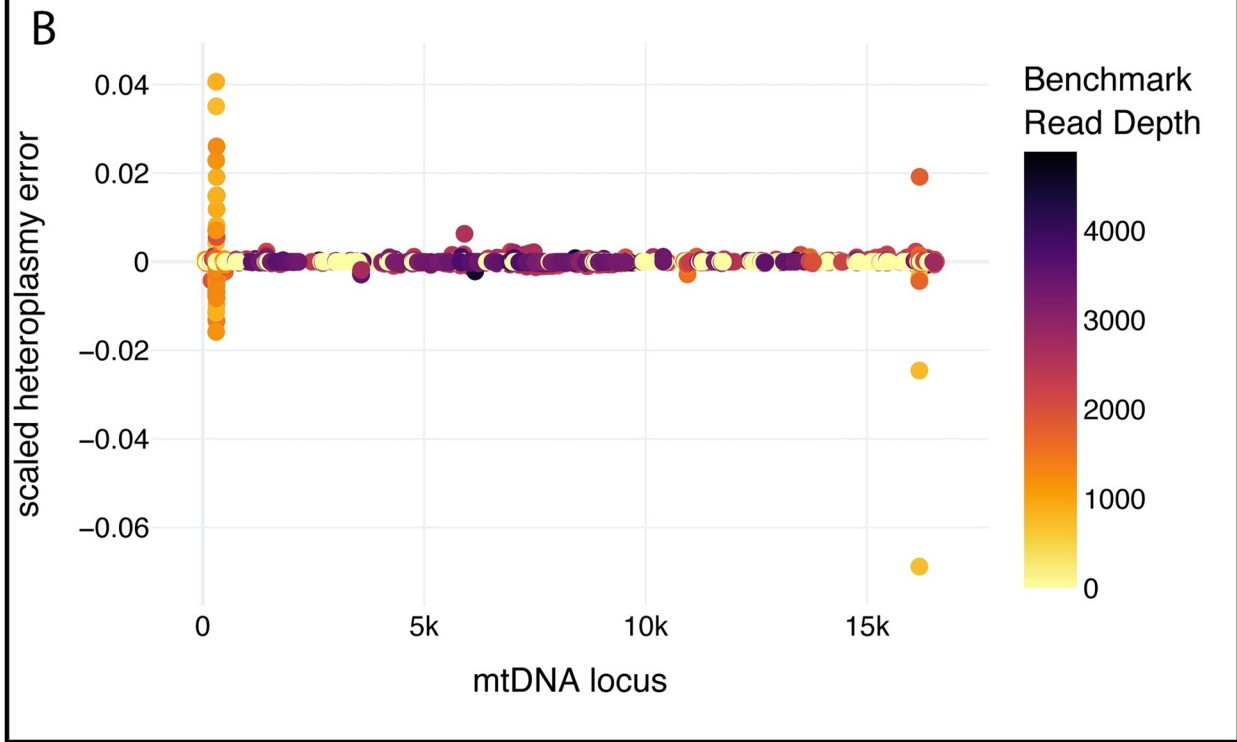

**Fig 3. Plot of heteroplasmy error between Benchmark variants and MitoScape variants, for each variant in each sample.** The x-axis represents the position in the rCRS. Benchmark read depth represents the read depth of the variant from the Benchmark dataset. Heteroplasmy error in a given sample and mtDNA locus is defined as the heteroplasmy value from the Benchmark variant set minus the heteroplasmy computed using MitoScape. Note that heteroplasmy error is a difference in fractions or percentages, not the percentage error. **A. Raw Heteroplasmy Error**. **B. Scaled Heteroplasmy Error**: Heteroplasmy error is scaled by $p(1-p)$ where p is the benchmark heteroplasmy.

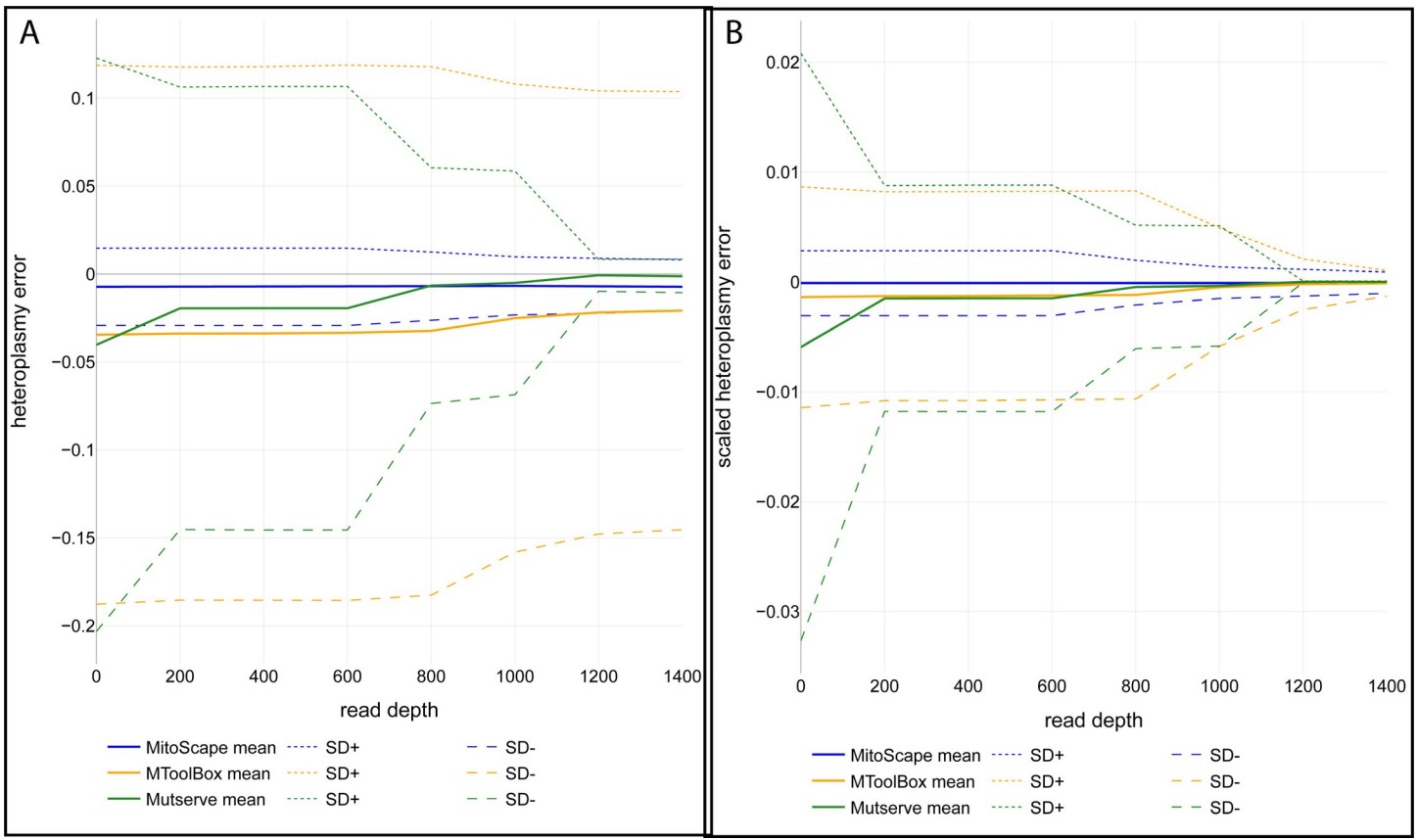

**Fig 4. Summary statistics of heteroplasmy error for MitoScape, MToolBox, and mtDNA-Server (Mutserve).** Heteroplasmy error in each sample and mtDNA locus is defined as the heteroplasmy value from the Benchmark variant set minus the heteroplasmy computed using MitoScape, MToolBox, or mtDNA-Server. **A. Raw Heteroplasmy Error**. **B. Scaled Heteroplasmy Error**: heteroplasmy error is scaled by p(1-p) where p is the benchmark heteroplasmy.

obtaining accurate heteroplasmy. Moreover, as the read depth of the WGS data increased, the standard deviation in heteroplasmy error decreased (Fig 4).

Mitochondrial DNA (mtDNA) copy number—defined as the ratio of mtDNA copies to nDNA copies—potentially impacts the effect of NUMTs on heteroplasmy detection. It is conceivable that the samples with lower mtDNA copy numbers would have greater error in mtDNA heteroplasmy levels. Hence, we examined the relationship between mtDNA copy number and MitoScape heteroplasmy error. We computed mtDNA copy number as the ratio of number of mtDNA reads to nDNA reads in chromosomes 1–22 for each of the nine test samples. We found no obvious correlation between mtDNA copy number and MitoScape heteroplasmy error (S5 Fig). These results are inconclusive, however, likely given the small N—nine samples—and the low amount of variation in the mtDNA copy number (mean = 90, standard deviation = 25).

## Comparison of MitoScape with standard tools

We compared MitoScape to two common tools for obtaining mtDNA variants from NGS data: MToolBox [18] and mtDNA-Server [21]. MToolBox is the standard tool used by the MSeqDR consortium, a large, global consortium for mitochondrial disease research consisting of a team of over 100 mitochondrial disease experts [33]. MtDNA-Server is a scalable NGS analysis workflow based on Apache Hadoop, for obtaining variants from human mtDNA data.

MtDNA-Server achieved similar or superior performance to several other computational tools for mtDNA analysis and ultra-sensitive variant detection [21]. To handle NUMTs, and low-complexity regions, mtDNA-Server adopts a post-filtering approach of tagging and filtering variants that are in these regions, or those variants that meet various thresholds. The schemes used in both MToolBox and mtDNA-Server are representative of all computational methods used to analyze mtDNA variants. We used MToolBox and the standalone version of mtDNA-Server—Mutserve v2.0.0rc10—to determine mtDNA variants from the same 22q11.2DS samples as used in the Benchmark and MitoScape mtDNA variant sets. We then determined the number of false negatives and false positive variants detected by both MitoScape, MToolBox, and mtDNA-Server, using the Benchmark variant set as our gold standard. Hence, the heteroplasmy values from the Benchmark variant set represents the actual heteroplasmy.

We next compared misclassifications in MitoScape, MToolBox, and mtDNA-Server. We defined any variant to be a false negative or missing if the heteroplasmy error is less than -0.2. In other words, we allow for the computed heteroplasmy to be 0.2 less than the Benchmark estimate of heteroplasmy but no more. For example, if the actual heteroplasmy is 0.5, then the corresponding heteroplasmy from the tested software would have to be greater than 0.3 for a match, and any heteroplasmy less than 0.3 is a false negative. We defined a false positive as any variant having an estimated heteroplasmy level greater than 0.2 plus the actual heteroplasmy. We used the same criteria for misclassifications to compare MitoScape, MToolBox, Mutect2, and mtDNA-Server. MitoScape was the most accurate in making heteroplasmic variant calls, having produced only one misclassification in the nine test samples as compared to 125 in MToolBox, and 21 in mtDNA-Server (Table 2). MitoScape's sole misclassification was a false negative, whereas the other two tools produced significant numbers of both false negatives and false positives, although mtDNA-Server was more accurate than MToolBox (Table 2 and S6 Fig). Surprisingly, both MToolBox produced errors in calling homoplasmic variants (defined as having heteroplasmy greater than 50%), with MToolBox and mtDNA-Server having 118 and 14 homoplasmic variant miscalls, respectively. MitoScape produced zero homoplasmic variant miscalls. The maximum absolute heteroplasmy error represents the maximum error in a heteroplasmy call made by each software. The minimum value for this measure by definition is zero in the best case, and one in the worst case. The maximum absolute heteroplasmy error produced by MitoScape was just 0.36 as compared to 1.0 for both MToolBox and mtDNA-Server (Table 2). Thus, MitoScape produced the most accurate heteroplasmy estimates in terms of every statistical measure amongst the three tools.

We also investigated the relationship between read depth and heteroplasmy error in calling heteroplasmic variants. We found that on average, the heteroplasmy error for MitoScape was consistently lower than MToolBox: -0.5 to -1% versus -1 to -4% (Fig 4). Also, the standard deviation of heteroplasmy error from both MToolBox and mtDNA-Server were greater than

**Table 2. Comparison of errors in variant calling among MitoScape, MToolBox, and mtDNA-Server.** False negatives are variants that are in the Benchmark mtDNA variant set but not in the corresponding tool (MitoScape, MToolBox, or mtDNA-Server) mtDNA variant set. Conversely, false positives are not in the Benchmark mtDNA variant set but were called by the corresponding tool (MitoScape, MToolBox, or mtDNA-Server). A variant is regarded as not detected if the heteroplasmy error exceeds 0.2. The maximum absolute heteroplasmy error ranges from 0.0 (best possible) to 1.0 (worst possible).

| Statistic | MitoScape | MToolBox | mtDNA-Server |
|---|---|---|---|
| Number of misclassifications | 1 | 125 | 21 |
| Number of false positives | 0 | 5 | 1 |
| Number of false negatives | 1 | 120 | 20 |
| Number of homoplasmic variant errors | 0 | 118 | 14 |
| Maximum absolute heteroplasmy error | 0.36 | 1.0 | 1.0 |

that of MitoScape, indicating that MToolBox and mtDNA-Server had variants with much larger errors in heteroplasmy levels than MitoScape. The differences in errors are most dramatic for low read depths, indicating that MitoScape is the best tool for calling variants with low read depths. These results also suggest that on average, MitoScape can reliably detect mtDNA variants as low as 0.005–0.01 heteroplasmy. The lower limit on average for MToolBox is at least three-fold higher at 0.03–0.04. Furthermore, these results demonstrate, however, that determining an absolute lower limit of detection of heteroplasmy is not possible, as the detection limit depends on the variant and read depth.

These results remained consistent if we used the scaled heteroplasmy error as opposed to the raw heteroplasmy error. MitoScape was the only tool for which the scaled heteroplasmy error was zero regardless of read depth, and the standard deviation of scaled heteroplasmy error was consistently lower than both MToolBox and Mutserve up to a read depth of approximately 1200. For read depths greater than 1200, the standard deviation of Mutserve was lower than both MitoScape and MToolBox. For all tools, however, the standard deviation of the scaled heteroplasmy error with read depths greater than 1200, is in (-0.001, 0.001) and is effectively zero being below measurement error. Thus, at read depths greater than 1200 the difference in performance of all tools is negligible. At read depths lower than 1200, MitoScape performs best in terms of heteroplasmy error, both raw and scaled.

While accurate heteroplasmy level measurement is critical in diagnosing mitochondrial disease [34], there are other important metrics for evaluating the performance of MitoScape and related tools. Accordingly, we sought to determine the detection error of benchmark mtDNA variants. We define detection error as the fraction of mtDNA variants from the benchmark dataset that can be detected by MitoScape and related software tools at different heteroplasmy thresholds. We found that MitoScape consistently detected a large fraction of mtDNA variants than either MToolBox or Mutserve for all minor allele frequency thresholds (Fig 5). Moreover, MitoScape detected almost 100% of the benchmark mtDNA variants that were above 0.1 heteroplasmy. In contrast, MToolBox detected at most half of the mtDNA variants regardless of heteroplasmy threshold, and Mutserve detected a maximum of 94% of the mtDNA variants.

## Application to complex human disease: Hypertrophic cardiomyopathy

We provide an example of applications of MitoScape to complex human disease where variants from both nuclear DNA and mtDNA are required. Our results have shown that both MToolBox and Mutserve can lead to erroneous homplasmic mtDNA variant calls (**Table 2**), which in turn potentially leads to erroneous mtDNA haplogroup calls. Thus, even for haplogroup calls, Mitoscape adds value and improved performance. Hypertrophic cardiomyopathy (HCM) is the most common genetic disorder of the heart, and is characterized by left ventricular hypertrophy [35]. HCM is thought to be associated primarily with mutations in 11 or more genes, but genotype-phenotype associations have been inconsistent suggesting that other genetic or environmental factors are at play. In particular, studies have shown an association between HCM and mitochondrial haplogroups in European populations [36,37]. We tested the association between HCM and mitochondrial haplogroups in an American population, using 7,184 whole exome sequences from the Penn Medicine Biobank (S1 and S2 Tables). These haplogroup calls were made based on the homoplasmic mtDNA variants calls generated from the MitoScape workflow. We found that men but not women in haplogroup T were at 3.52 times higher risk for HCM than those in the most common European haplogroup, R0 (S3 Table, adjusted p-value = 0.003), thus corroborating reported associations [36]. In contrast to the Castro et al study [36], MitoScape produces the complete mtDNA sequence as opposed to a small subset of single nucleotide changes, and hence more precise mitochondrial haplogroup information.

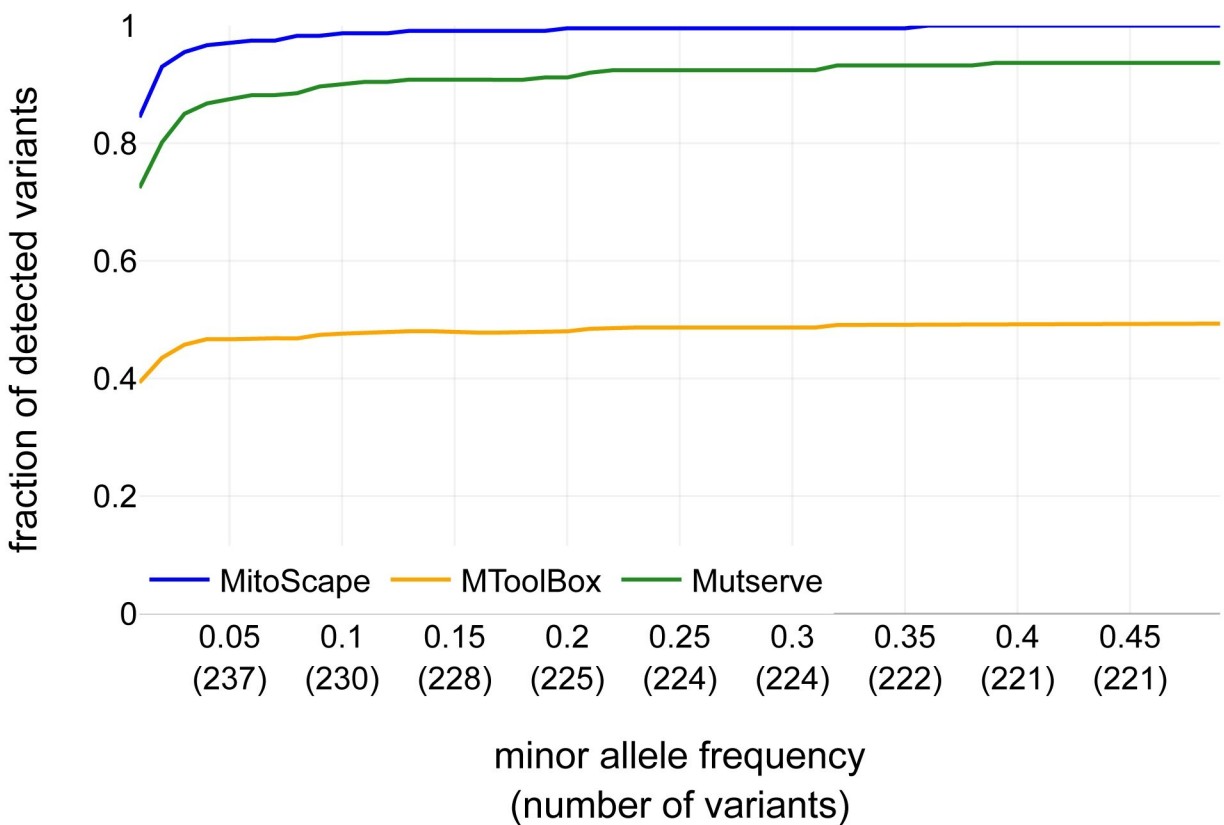

**Fig 5.** Comparison of the fraction of benchmark variants detected (y-axis) versus the heteroplasmy threshold for detection (x-axis), for the MitoScape, MToolBox, and Mutserve. The number of heteroplasmic mtDNA variants is shown in parentheses.

## Summary

We have presented MitoScape, a novel, machine learning-based software to align mtDNA from NGS data. We have also demonstrated the superior performance of MitoScape compared to two common tools for obtaining mtDNA variants from NGS. MToolBox and mtDNA-Server produce 125- and 21-fold more misclassifications than MitoScape. Importantly, MitoScape has several additional advantages over post-filtering approaches as described in the Introduction. First, alignment tools, variant callers and sequencing technology are all likely to improve over time. Unlike other tools, the design of MitoScape allows for these components to be changed without modification to the software. For instance, different sequence aligners and variant callers can be readily used in the MitoScape framework depending on the research problem. Second, MitoScape has the ability to attenuate the prediction probability to allow for varying the percentage of false positives and negatives based on the user's needs—a powerful feature that is unique to MitoScape. Third, with MitoScape the resulting classified sequence alignments can be used to determine mtDNA copy number. MtDNA copy number is an important source of mitochondrial variation and plays an important role in the pathophysiology of certain human diseases, especially mtDNA depletion syndromes [38]. Computing mtDNA copy number is not possible in post-filtering approaches such as mtDNA-Server.

Obviously, an important goal of mitochondrial genomics is to identify mtDNA variation, including mtDNA copy number, which alter mitochondrial function. MitoScape goes beyond other software by not only accurately identifying mtDNA sequences, but also modeling

mitochondrial genetics through machine learning. Unique alignment and post-filtering approaches do not and cannot model salient aspects of mitochondrial biology. MitoScape takes a novel departure in identifying mtDNA sequence by adopting for the first time, rho-zero cells modeling NGS sequencing of NUMTs. MitoScape also incorporates a highly flexible framework allowing for different NUMTs and even training sets to be modified to account for mtDNA variation in different tissues, diseases and populations. For instance, if we are studying cancer, using both rho-zero cells and mtDNA-enriched data from cancer cells will provide more accurate estimates of heteroplasmy that general approaches. This flexible approach guards against over-fitting and permits analysis in a context-specific manner which is critical for studying mitochondrial genetics. No other software offers this flexibility. Another advantage of our novel approach is that once new variants are discovered and more training data is produced, these data can be used to continually update and improve classification. These design choices allow for obtaining high precision and accurate mtDNA variants from NGS data, which will be vital in the diagnosis of both primary mitochondrial and complex human disease.

## Availability and future directions

**Project name**: MitoScape.

**Project home page**: https://cavatica.sbgenomics.com/public/apps#d3b-bixu/app-publisher/mitoscape-wf/ including full instructions on how to run MitoScape on the Seven Bridges Cavatica platform.

**Source home page:** https://github.com/larryns/MitoScape.

**Operating System:** Platform independent.

**Programming Language**: Scala.

Data specific to HCM analysis are available from the Penn Medicine Biobank (https://pmbb.med.upenn.edu/). All other data, including Benchmark data, are available via authorized access from https://cavatica.sbgenomics.com/u/cavatica/22q11-deletion-syndrome-project/.

A strength of MitoScape is the availability to add more data including positive (mtDNA-enriched) data and negative (rho-zero) data. Additional data in the form of NUMT locations can also be added to improve, adapt or extend MitoScape to specific datasets, including non-human data.

## Supporting information

**S1 Supplementary Methods. Details of supplementary methods not covered in main text.** (DOCX)

**S1 Fig.** Plot of relative feature (variable) importance scores (y-axis) after training of random forest classifier in MitoScape. Each feature is displayed on the x-axis. Feature importance scores of all variables sums to one, and the higher the relative variable importance score, the more important this feature was in the classification procedure. (TIF)

**S2 Fig. Summary of unaligned sequencing reads from nine 22QDS samples to both rCRS and RSRS.** (TIF)

**S3 Fig. Summary of reads aligning to mitochondrial DNA reference (rCRS or RSRS) from nine 22QDS samples.** (TIF)

**S4 Fig. Summary statistics of number and frequency of heteroplasmic mtDNA variants identified in the 9 benchmark test data samples.**
(TIF)

**S5 Fig. Relationship between mitochondrial DNA copy number and heteroplasmy error from MitoScape. The red diamonds represent the mean heteroplasmy error for a given mitochondrial copy number.** Each blue circle represents the heteroplasmy error and mitochondrial copy number for a single variant.
(TIF)

**S6 Fig. Summary of false negative misclassifications for MitoScape, MToolBox, and Mutserve (mtDNA-Server).** The y-axis represents the cumulative number of false negatives where the corresponding actual heteroplasmy is less than the value on the x-axis. The x-axis represents minor allele frequency, and therefore, is between 0 and 0.5. Minor allele frequency is equal to actual heteroplasmy if actual heteroplasmy is < 0.5, and equal to 1-actual heteroplasmy, otherwise.
(TIF)

**S1 Table. Penn Medicine Biobank Participant Characteristics.**
(DOCX)

**S2 Table. Haplogroup demographics of subjects used in hypertrophic cardiomyopathy-mitochondrial haplogroup association from Penn Biobank data.**
(DOCX)

**S3 Table. Logistic regression analysis with HCM as dependent variable and mitochondrial haplogroups, age, and the first five principal components of the nuclear genetic variants PCA analysis as covariates, for men only.** Reference haplogroup is R0. Adjustment for multiple testing was done by Bonferroni correction. Logistic regression was performed using R.
(DOCX)

## Acknowledgments

The authors thank the Regeneron Genetics Center for supplying sequencing and genetic variant data on hypertrophic cardiomyopathy.

## Author Contributions

**Conceptualization:** Larry N. Singh.

**Data curation:** Larry N. Singh.

**Formal analysis:** Larry N. Singh, Noah L. Tsao, Scott M. Damrauer.

**Funding acquisition:** Stewart A. Anderson, Douglas C. Wallace.

**Investigation:** Larry N. Singh, Bryn Loneragan, M. Isabel G. Lopez Sanchez, Jianping Li, Patrick Acheampong, Ian A. Trounce.

**Methodology:** Larry N. Singh.

**Project administration:** Larry N. Singh.

**Resources:** Oanh Tran, Beverly S. Emanuel, Daniel J. Rader, Zoltan Arany, Adam C. Resnick, Stewart A. Anderson, Douglas C. Wallace.

**Software:** Larry N. Singh, Brian Ennis, Yuankun Zhu.

**Supervision:** Larry N. Singh, Ian A. Trounce, Prasanth Potluri, Zoltan Arany, Scott M. Damrauer, Stewart A. Anderson.

**Validation:** Larry N. Singh.

**Visualization:** Larry N. Singh.

**Writing – original draft:** Larry N. Singh.

**Writing – review & editing:** Larry N. Singh.

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
