## [Decision Letter · Decision Letter 0]

24 Jul 2021

Dear Dr.. Singh,

Thank you very much for submitting your manuscript "MitoScape: A big-data, machine-learning platform for obtaining mitochondrial DNA from next-generation sequencing data" for consideration at PLOS Computational Biology. As with all papers reviewed by the journal, your manuscript was reviewed by members of the editorial board and by several independent reviewers. The reviewers appreciated the attention to an important topic. Based on the reviews, we are likely to accept this manuscript for publication, providing that you modify the manuscript according to the review recommendations.

Sincerely,

Manja Marz

Software Editor

PLOS Computational Biology

Manja Marz

Software Editor

PLOS Computational Biology

[LINK]

Reviewer's Responses to Questions

**Comments to the Authors:**

Reviewer #1: Comments on “MitoScape: A big-data, machine-learning platform for obtaining mitochondrial DNA from next-generation sequencing data”

Larry N. Singh et al. present in this manuscript a novel software, MitoScape, developed for analyzing human mitochondrial DNA. MitoScape leverages mtDNA reads produced from next-generation sequencing experiments to infer mtDNA variants, including homoplasmies and heteroplasmies, and has the potential to assess mtDNA copy numbers. An elegant design of MitoScape is that it can accurately classify reads from mtDNA and reads from NUMTS based on a random forest classifier trained using features of alignments from mtDNA-enriched libraries and alignments from mtDNA-depleted libraries. The authors show significantly improved accuracy of MitoScape in identifying mtDNA heteroplasmies and homoplasmies as compared to two other mtDNA analytical tools, MToolBox, and mtDNA-Server. The authors further exemplify the use of MitoScape in a clinical study of mtDNA which enables inference of the full-length human mtDNA sequence.

The manuscript on MitoScape is well-written, accompanied with detailed online instruction on how to integrate MitoScape into the Seven Bridges Cavatica platform, a secure, cloud-based environment for analyzing large-scale genomics data. The source code of MitoScape is written in scala which has been made available on GitHub.

In light of increasing availability of high-quality next-generation sequencing data, an improved analytical tool like MitoScape will definitely accelerate the discovery of mtDNA’s roles in human diseases. Thus, I recommend this study for publication in PLOS Computational Biology.

I only have one minor suggestion.

An important factor influencing mtDNA variant detection is mtDNA copy number. Normally, samples with lower mtDNA copy numbers would be more significantly affected by reads from NUMTS. I wonder what distribution of mtDNA copy numbers in the nine testing blood samples is, given the corresponding data were generated by WGS. Does mtDNA copy number affect performance of mtDNA variant identification, especially identification of heteroplasmies? Providing copy number data could also illustrate another important use of MitoScape, as mentioned in Discussion.

Reviewer #2: The accurate identification of mitochondrial DNA (mtDNA) variants, especially low-frequency heteroplasmies, in sequencing data is an important problem in mitochondrial genetics. This problem is especially acute due to the presence of nuclear DNA fragments masquerading as mitochondrial DNA in sequencing data, commonly known as “NUMTs”. There are many approaches available to minimize NUMT contamination in sequencing data. However, these are largely heuristic, rely on the use of arbitrary filters, and there is often no clear measure of their efficacy/accuracy. Singh et al. provide an effective, scalable method to deal with these problems. I have a number of questions and comments to which I would like to hear the authors’ response:

The authors mentioned 10 samples enriched for mtDNA sequences as positive control. Am I correct in thinking that these independent from the 9 samples used as the test set? Perhaps make this clearer in the main text.

They mention using k-fold cross-validation with 80% of the training set and 20% as validation. Could the authors describe this in more detail? There are 8 samples in the negative training set and 10 in the positive set. How were they split? Or was the separation into training and validation sets carried out on the read level? How many iterations of the k-fold validation were carried out? Could the authors provide performance metrics from the k-fold validations? Can they also provide the weights that the classifier assigns to each of the features used? Additionally, can they also generate a plot of the probability of correctly assigning a read to mt/NUMT sequences as a function of these features in their validation? These metrics will be useful in understanding the classifier and how important each is in discriminating mitochondrial and NUMT sequences.

I am not sure using the absolute difference in heteroplasmy frequency between the benchmark and test set is the appropriate way to measure the performance of the method. Higher frequency heteroplasmies are more likely to vary in frequency just by sampling error. Perhaps more accurate would be to scale the difference by p(1-p) where p is the heteroplasmy fraction in the benchmark dataset. It would also be helpful to see this difference with respect to the frequency of the heteroplasmy. Please also provide similar metrics or a figure (e.g. as in Fig 3) for the other methods for a direct comparison with Mitoscape.

I noticed in Fig. 4 that Mutserve matches or performs better than Mitoscape at higher sequencing depth (>1200). Would this be a fair assessment? If so, perhaps mention this as it does not take away from the fact that Mitoscape performs well at lower sequencing depths.

I do think an absolute difference of 0.2 is appropriate to define false negative or positive heteroplasmies. One reason for this is that a difference of 0.2 matters means different things at low and high frequency heteroplasmies. For example, at a frequency of 0.2, a difference of -0.2 genuinely indicates a failure to detect the heteroplasmy whereas the same difference at a frequency of 0.5 does not. Additionally, I feel that heteroplasmy classification errors (as defined here) should be measured by detection (or failure of) heteroplasmies, and not based on difference in frequency of known heteroplasmies. For example, a false positive could be detection of a heteroplasmy above some frequency threshold (e.g. 0.05) which did not exist in the benchmark data.

I did not understand the purpose of the section “Application to complex human disease: hypertrophic cardiomyopathy”. Was Mitoscape used to call mtDNA reads for haplogroup calling? How much more accurate was this process compared to just calling haplogroups from the sequence data (without any filters) using Haplogrep (for example)? Alternatively, how much more accurate was Mitoscape compared to other methods described in the paper? In other words, I see the benefit of using Mitoscape to call low-frequency heteroplasmies, but I don’t see how beneficial it is for haplogroup calling.

Please provide details of the number and frequency distribution of heteroplasmies identified in the 9 test samples.

**Have the authors made all data and (if applicable) computational code underlying the findings in their manuscript fully available?**

Reviewer #1: Yes

Reviewer #2: Yes

PLOS authors have the option to publish the peer review history of their article (what does this mean?). If published, this will include your full peer review and any attached files.

Reviewer #1: No

Reviewer #2: No

Figure Files:

Data Requirements:

Reproducibility:

References:

---

## [Decision Letter · Decision Letter 1]

27 Oct 2021

Dear Dr.. Singh,

We are pleased to inform you that your manuscript 'MitoScape: A big-data, machine-learning platform for obtaining mitochondrial DNA from next-generation sequencing data' has been provisionally accepted for publication in PLOS Computational Biology.

Best regards,

Manja Marz

Software Editor

PLOS Computational Biology

Jason A. Papin

Editor-in-Chief

PLOS Computational Biology

Reviewer's Responses to Questions

**Comments to the Authors:**

Reviewer #2: I see the authors point about how Mitoscape improves homoplasmic variants, which could lead to improvement in haplogroup calling. However, I still find the section "Application to complex human disease: hypertrophic cardiomyopathy" odd because it does not test the accuracy of haplogroup calling directly. Instead, it discusses an association result, which is a downstream application but is not very relevant to the performance of Mitoscape. I understand the authors argument about not having haplogroup calls from the other tools to compare. I think this may be a personal choice and I will leave the decision to keep this section or move to the supplement to the authors.

The authors satisfactorily addressed my questions and I have no further questions or comments.

**Have the authors made all data and (if applicable) computational code underlying the findings in their manuscript fully available?**

Reviewer #2: Yes

PLOS authors have the option to publish the peer review history of their article (what does this mean?). If published, this will include your full peer review and any attached files.

Reviewer #2: No

---

## [Editor Report · Acceptance letter]

5 Nov 2021

PCOMPBIOL-D-21-00835R1 

MitoScape: A big-data, machine-learning platform for obtaining mitochondrial DNA from next-generation sequencing data

Dear Dr Singh,

I am pleased to inform you that your manuscript has been formally accepted for publication in PLOS Computational Biology. Your manuscript is now with our production department and you will be notified of the publication date in due course.

With kind regards,

Olena Szabo
